# Efforts toward Creating a Sustainable Business Model: An Empirical Investigation of Small-Scale Certified Forestry Firms in Taiwan

**Jun-Yen Lee and Ching-Hsing Chang \***

Department of BioBusiness Management, National Chiayi University, Chiayi City 600, Taiwan;
leejy@mail.ncyu.edu.tw
\* Correspondence: chc@mail.ncyu.edu.tw; Tel.: +886-5273-2878

**Abstract:** Forest certification plays a notable role in promoting sustainability. This certification shows that forestland holders have adopted innovative practices toward realizing sustainable business models. Relatively little analysis has been devoted to identifying the efforts of transforming a conventional business model into a sustainable version through the application of forest certification. This paper examines the elements of a sustainable business model: value proposition, value creation and delivery, and value capture in certified forestland holders' business operations. Empirical results have confirmed that certification signifies a successful sustainability transformation within adopting firms. However, these small organizations struggle with obtaining know-how regarding sustainable forest management. There needs to be adequate external support, such as government consultants or academic researchers, in order to successfully adopt third-party forest certification. However, while practicing sustainable forest management activities will not guarantee premium prices, the certification has, in some rare cases, helped to develop a new niche market. Good communication with stakeholders has improved firms' relationships with local residents, but more channels of communication are still needed to activate green consumers.

**Keywords:** sustainable business model; sustainable forest management; forest certification; business model innovation

## 1. Introduction

The Earth Summit, a world summit on sustainable development held in Rio de Janeiro in 1992, emphasized the statement that sustainable development counters the challenges caused by environmental degradation and social inequality [1]. The summit accentuated the importance of not compromising the developmental and environmental needs of future generations with current economic prosperity. In the United Nations' 2030 Agenda, sustainable forest management is one of seventeen sustainable development goals (SDGs). Currently, there are two major approaches that can be adopted to pursue agreement among three seemingly conflicting goals of forest management: environmental protection, social equality, and economic prosperity. In the first approach, the Food and Agriculture Organization of United States (FAO) has developed a toolbox to facilitate the adoption of sustainable forest management (SFM) [2]. Building upon the four pillars of sustainability (environmental, social, cultural, and economic), the SFM toolbox aims to safeguard the longevity of forests while still supplying the diverse needs of society by providing tools, case studies, and other resources for sustainable forest management, and by organizing these into modules to improve accessibility for forestry firms, managers, and other stakeholders. Forest certification provides another option where a third party (certifier or certification body) monitors the process of certification and ensures the sustainability of

managerial practices [3–5]. By adhering to its principles and criteria, forest certification promotes SFM within a context of pre-assessing, planning, implementing, and monitoring processes.

Two prominent forest certification systems, the Programme for the Endorsement of Forest Certification (PEFC) and the Forest Stewardship Council (FSC), promote environmentally appropriate, socially beneficial, and economically viable forest management. PEFC, an international non-profit, non-governmental organization established by 11 national organizations, has endorsed 43 national certification systems and certified 310 million hectares of forests throughout the world since its inception in 1999 [6]. The FSC is regarded as a voluntary, non-governmental, and market-driven stewardship system. It was initiated in 1993 and developed by a coalition of international environmental NGOs, social, and economic groups [7,8]. It highlights the importance of balancing economic viability, social benefits, and environmental quality [9]. It has certified approximately 200 million hectares of forestlands [10]. As the most pervasive of the programs, the FSC introduced a unified and global forest management certification standard for responsibly managed forestlands.

Previous research has proven that forest certification plays a vital role in promoting sustainable forest management [11]. Forest owners who join forest certification programs perceive that their forestry practices lead to improved environmental practices [12,13]. However, according to a survey that focused on forest managers and related stakeholders of industrial plantations in Spain, less than 20% of forest managers and experts agree with the statement that "Plantations are certified to guarantee their sustainability" [14]. Klinberg (2003) suggests that certification may be perceived by forest owners only as a tool to communicate with consumers [15]. Others suggest that the motives to become certified can vary from gaining public trust, improving company image [16–19], and building relationships with stakeholders [15,20,21] to increasing economic return [22].

Although only a few studies have recognized that the pursuit of sustainability is a major driver for certificate adoption, research has proven that forest certification plays a key role in promoting the sustainability of managed forests and of forestry [11,23]. Being certified by PEFC or FSC requires firms to practice forest management in which the values of environmental protection, social benefits, and economic performance are simultaneously taken into consideration. It can be inferred that forest certification can act as a catalyst for forest firms to change their business models from conventional to sustainable. For instance, FSC's principles and criteria help forest firms to modify their value proposition to focus not only on customer interest, but also on the wellbeing of all stakeholders. In so doing, value creation and delivery is thus expanded to deliver a positive environmental and social impact. We posit that forest certification provides an ideal means to initiate a change to sustainability by offering technical know-how, and that by practicing these procedures as required by certification, it offers an opportunity for forest owners to understand and embrace the idea of sustainable development.

Business model innovation re-conceptualizes a firm's purpose and initiates a change in value creation [24]. Therefore, it is an indispensable step in systematically and continuously converting businesses models to sustainability [25]. Previous studies suggest that such transformations can be undertaken with either an inside-out or an outside-in approach. Although transformation by either of these approaches may take different routes, their effectiveness has been validated by several empirical studies [26–29]. An inside-out approach can be initiated by examining a firm's current operation with a triple layered business model canvas (TLBMC), with which a firm explores potential market segments and modifies processes to fit a sustainability-driven business model [28]. TLBMC expands Osterwalder & Pigneur's (2010) business model canvas by explicitly integrating environmental and social value creation into the former model. It helps to identify potential concerns in the current operation and is a tool used by socially conscious firms to shift their conventional business models to sustainable ones. However, the adoption of this approach appears to be difficult for resource-constrained firms [30]. A well-developed management system and even a R&D department are required to facilitate such a transformation [28]. In contrast, an outside-in approach involves seeking external insights, specifically in seeking knowledge on the processes of business model innovation. These insights

can be gained through learning from idealized business model archetypes as developed by other organizations in creating guidelines for model innovation [26]. This approach is ideal for small firms that lack adequate resources or knowledge to initiate their own innovative practices.

Adopting forest certification is considered an archetype of the outside-in approach that forest firms can resort to when carrying out their business model innovation toward sustainability. However, earlier forest management literature lacks significant insights into the transformation process and how firms' activities, such as pre-assessing, planning, implementing, and monitoring, need to change to align with these sustainability-driven business models. Wells and Nieuwenhuis (2004) indicate that the architecture, principles, and components of a firm's business model need to be evaluated in order to understand the results of business model innovation [31].

The contribution of this study is twofold. As an effort to create a theoretical causal relationship between FSC adoption and sustainability transformation, we link Joyce and Paquin's (2016) business model to the principles and criteria of FSC certification, and gauge whether all three layers of the model are considered by the certificate [28]. The authors also show that FSC can be categorized into one of the idealized archetypes of a sustainable business model, the stewardship role, an outside-in approach to initiate business model innovation [24]. Meanwhile, this study fills the gaps in existing literature insofar as it takes an insightful look at how small-scale forest firms, having adopted FSC, transform their conventional business models to ones of sustainability, and to what extent certified forestland holders have taken sustainability completely to heart. The challenges they encounter during business model innovation are very important, and practitioners, or perhaps government agencies, should work on solutions to ease the transformation to sustainability.

Specifically, the authors examine the following questions: Have the product and service values changed for customers as a result of forest certification? How are values created and delivered? Have management activities changed, in terms of pre-assessing, planning, implementing, and monitoring? What are the motives and challenges borne by forestry firms when implementing FSC? To answer these questions, this study carried out exploratory research and developed semi-structured questions drawing from prior literature on sustainable business models, and targeted small-forest managers in Taiwan who had recently been FSC certified. This discourse is structured as follows: Section 2 explores the elements of sustainable business models and links them to FSC adoption; Section 3 presents the background of certified Taiwanese forest firms; Section 4 deals with the questionnaire design and sample selection; Section 5 summarizes the changes and challenges encountered during the sustainability transformation; and Section 6 details conclusions drawn.

## 2. Sustainable Business Model Elements for Small-Scale Forestry Firms

Changing environmental factors, such as climate change, resource shortages, and rapid technological progress highlight the inadequacy of traditional management practices for meeting these future challenges. Conventional business models are market-oriented and focus on delivering product or service values to customers. In contrast, sustainable business models deliver product or service values not only to customers, but also to all stakeholders. Amit and Zott (2012) state that these models change 'the way you do business' rather than 'what you do' [32]. They go beyond process and product inventions, and focus on developing new management systems to meet the needs of sustainable development [33]. Sustainable business models build on the triple bottom line approach, related to economic, social, and environmental aspects of the market and society. The models not only aim to maximize firms' performances in economic profit, but also to deliver social and environmental benefits [25]. Beyond the traditional customer focus, they change firms' value-network perceptions to cover a wider set of stakeholders, which in turn triggers innovation and transformation of the business models [34,35]. Boons and Lüdeke-Freund (2013) propose the following normative requirements for sustainable innovations: the value proposition must include environmental, social, and economic value; the business infrastructure must incorporate sustainable supply chain management; the customer interface must cover not only customers, but other stakeholders (value creation and delivery);

the financial model should consider the economic costs and benefits of all involved players (Value capture) [26]. Building on the concept of Osterwalder and Pigneur's (2010) business model elements, Bocken et al. (2014) define sustainable business model archetypes based on the different activities undertaken in relation to three major business elements: value proposition and the consideration of product/service, customer segments, and stakeholder relationships; value creation and delivery and its incorporation of key operational activities, resources, channels, partners, and technology; and value capture that includes cost structures and revenue streams [24,30]. The authors categorized sustainable business models into three groups; technological, social, and organizational. Significantly, the social archetype group follows the "adopt a stewardship role" archetype by proactively engaging with all stakeholders to ensure their long-term health and wellbeing. A value proposition for this archetype will embed environmental and social benefits in the products and services provided. Value creation and delivery aligns firms' activities and partners to realize these benefits, and furthers the goal of long-term economic benefit and/or reputation.

One may ask if there is a strong linkage between forest certification and sustainability, and if a sustainable business model is a fit for certified forest land holders. Gullison (2003) observed an improvement in the value of biodiversity after a change in management practices as a result of FSC adoption [36]. In the same vein, previous research regarding forest certification has found that certification is an important instrument of change in the promotion of sustainable forest management [22,23,37–39]. In addition, the value of sustainability may also be internalized within the mindsets of managers during its adoption. Empirical evidence suggests that certified forestry firms perceive that their forest practices lead to an improved environment [12,13]. Another advantage enjoyed by FSC is that it can be replicated, a quality which enables the sustainable transformation of markets [40]. FSC is undertaking modular approach to promote the adoption of its standards. Any interested organization can access its information online and contact the certifying body worldwide to initiate adoption. Such practices help to push FSC products into mass markets while promoting sustainable forest management.

This study postulates that participating in forest certification will not only affect small firms' forestry management practices, but will also initiate engagements with their stakeholders in the pursuance of resolving social and environmental impacts. To further comprehend this process, the authors propose a sustainable business model for forestry firms, which builds on the "adopt a stewardship role" archetype as outlined by Bocken et al. (2014) (see Table 1) [24].

**Table 1.** A sustainable business model for forestry firms and its relation to previous models.

| Models | Model Elements | | | Sources |
| --- | --- | --- | --- | --- |
| | Value Proposition | Value Creation and Delivery | Value Capture | |
| Conventional business model | Product/service offering to satisfy customer and generate firm's economic return | Firm's activities, resources, technology, channels, partners, etc. | Concerning the economic profit from product/service provisions. | Adopted from [30] |
| 'Adopt a stewardship role' archetype | Bring broader benefits to stakeholders (including customers, workers, communities, etc.) by providing tangible products and intangible services at minimum social costs. | Ensuring firms' and its partners' activities create and deliver environmental and social benefits to customers. | It may generate brand value and long-term business benefits from stakeholders' wellbeing and health. | Adopted from [24] |
| Adopt FSC certification | Providing certified timber and non-timber products/ecosystem services with broader benefits (social and environmental values) to stakeholders and community. | Engaging with stakeholders to review forestry activities comprehensively to ensure the creating and delivering of products or services satisfying the market demand for social equality, environmental protection, and economic return. | Besides economic value, environmental and social values are also captured by ecosystem services' protection, landscape conservation, and by helping local community development, etc. | Organized by this study |

Source: organized by this study.

Value proposition, the first element, is defined as " ... the bundle of products and services that create value for a specific customer segment" [30,41]. The bundle of products and services can be any combination of tangible products/services and intangibles such as conserving biodiversity, ecosystem service, and helping the local community's economic development. In a sustainability-oriented firm, customers are not only buyers of the goods and services, but are also part of other stakeholder groups such as employees, community, environmental/social groups, etc. Satisfying the needs of all customer segments will bring long-term economic benefits. In the context of forest management, forest certification allows firms to practice responsible forestland management by offering timber/non-timber products while aiming to consciously and carefully handle environmental and social impacts. This is consistent with the ideas of sustainable value proposition.

The second element, value creation and delivery, involves a firm's resources, activities, partnerships, and channels. Resources can be facilities, financing, patents, human resources, etc. Activities executed by companies create value [42]. Disparate companies have distinctive activities offering diverse value creation and delivery. Typically, manufacturing firms' activities generate product and market values. The value creation of consultancies, law firms, hospitals, and other service firms is dependent upon their abilities to resolve the problems of their customers. For certified forest firms, the provision of timber/non-timber products and services not only focuses on conventional supply chain management, but also on the management of environmental and social issues [26,28]. Traditionally, companies build ties with suppliers and/or outsourcers. Sustainable value creation and delivery defines partnership as establishing relationships with communities, suppliers, and/or outsourcers [38,41]. Developing and maintaining long-term mutually beneficial relationships requires actively engaging with these stakeholders to understand their needs, prevent negative social impacts, and help with local economic development. As a result, the critical component in the successful sustainability transformation of certified forest firms is human expertise. Paramount is the capability of employees to identify and evaluate environmental and social issues influenced by the planned activities that will be carried out in forestlands. A discussion regarding the lack of human expertise in Taiwanese forest firms will be presented in Section 5.

Sustainable entrepreneurs can develop a niche market by introducing a new product variation that appeals to 'green' consumers. However, in light of evolutionary economics, considerable effort needs to be devoted to this new variation for it to survive the selection process before enjoying its market share expansion [40]. In order for this to happen, channels of communication, an important factor in value creation and delivery, should be utilized to educate customers, gain stakeholders' support, and reach potential buyers. The supply of information, such as details of management planning and the monitoring of results, delivered via internet or personal contacts, will ensure that a firm's efforts towards creating environmental and social values are conveyed to interested and affected stakeholders. At the same time, distribution networks ought to be expanded to increase the accessibility of new products. In this aspect, forest certification NGOs constantly update certified firms' operational activities on their websites and occasionally carry out web sales; this updating provides pertinent information to interested stakeholders and potential customers.

The construct of the third or last element, value capture, consists of economic, environmental, and social values. Economic value capture considers revenues generated from customers and costs incurred in producing products and services [41]. Environmental and social values are measured in terms of impacts and benefits. Ecological costs assess the environmental impacts, while social costs focus on the community impacts [28]. Environmental and social benefits address the positive impacts of forest activities.

The sustainable TLBMC business model can be applied to assess environmental impacts from firms' activities using a life-cycle-based environmental layer, while social layers provide guidance for preventing issues such as social inequality from arising. This comprehensive model takes an inside-out approach, whereby firm managers are encouraged to screen their business operations and watch for potential issues as outlined by the TLBMC model, as well as to monitor applicable inventions

to their current business models. On the other hand, the adoption of FSC can be regarded as an outside-in approach, whereby forestry firms can align themselves with the TLBMC business model, in that the development of FSC matches the triple bottom line perspective. Table 2 describes some principles and criteria of forest certification standards and their linkage to each individual element of the triple bottom lines. In summary, the forest certification standard computes ecological costs by evaluating the degrading ecosystem services, diminishing biodiversity, landscape destruction, etc., all of which can be mitigated or prevented using proactive measures. Social costs include workers' health and safety programs, grievance resolutions, compensation for loss and damages, and violation of intellectual property rights. When workers' rights, gender equality, minimum wages, training, rights of communities and indigenous peoples, provision of employment, and other services are taken into management consideration, social benefits are improved.

**Table 2.** Principles and criteria of the forest certification standard (FSC) related to three elements of sustainability.

| Elements of Sustainability | Principles and Criteria | Descriptions and Desired Outcomes in Forest Management |
|---|---|---|
| Environmental | Environmental value | Prevention, mitigation and repair of negative impact. Protection of rare and threatened species. Protection and restoration of native Ecosystems. Biodiversity protection. Water protection. Landscape. |
| Social | Workers' Rights | Workers' Rights at work. Gender equality. Health and safety. Minimum wages. Job specific training. Grievance resolution and compensation for loss or damage. |
| | Communities and Indigenous Peoples' Rights | Rights of communities & indigenous peoples. Management and protection of sites of special significance. Protection, utilization and compensation for traditional knowledge. Provision of employment, training and other services. Social and economic development. |
| Economic | | Production of diversified benefits and products. Sustained harvesting levels. Externalities. Local processing, services, and value adding. Long term economic viability. |

Source: organized by this study.

For conventional forestry firms, a business model innovation for sustainability converts the "disvalue" of externalities from conventional manufacturing processes into a strong sustainability quality that is valued by environmentally and socially conscious customers [43]. However, such a transformation is not without its challenges. Previous studies have identified the difficulties encountered in adopting a sustainable business model. Anecdotal evidence from ISO 14001 certification suggests that the main reason firms do not apply for certification is because of capital constraint. This is especially true for older firms where major expenses would have to be incurred in order to replace outdated equipment [44]. For small-scale foresters, the upholding of complex standards is the major obstacle that inhibits their adoption willingness. Another obstacle is that firms must devote a considerable amount of time and make use of a huge number of resources to obtain the certification, while consumer acceptance for certified products remains unknown [15].

Parrish (2010) stresses that a good grasp of "perpetual reasoning" is critical for achieving successful conversion. Contrary to the conventional wisdom of profit maximization, managers should switch their mindsets to balance the wellbeing for all stakeholders, instead of simply seeking to achieve economic efficiency [45]. They should manage with quality rather than quantity, and by the principle of worth contribution. The following sections will discuss the challenges faced by forestry firms that have orientated themselves to become social entrepreneurs by adopting FSC standards. The analysis will illustrate changes in the aspects of value proposition, value creation and delivery, and value capture, as well as the challenges following these changes. In particular, the difficulties arising from

pre-assessing, planning, implementing, and monitoring will be carefully addressed to help future FSC adopters quickly deal with the processes of transformation.

## 3. Background of Forestry Firms Adopting Forest Certification in Taiwan

Taiwan roundwood production dropped sharply from 74,188 $m^3$ in 1991 to 29,870 $m^3$ in 2015, averaging around 30,000 $m^3$, while the domestic demand is at 5 million $m^3$. Pressures from domestic environmental groups, the production quota (maximum of 200,000 $m^3$ annually) from the Taiwan Forestry Bureau, and other government regulations on log production threaten this industry. The worst threat is from environmental groups that consider logging an "evil activity." Forestry firms try to deal with this negativity by adopting third-party forest certifications recognized around the world. In so doing, these firms believe that the public will perceive certified forestlands as a sign of good forestry practice [13].

FSC forest certification is still in its infancy in Taiwan. In 2016, there were only five small FSC certified forestlands, ranging from 28.48 to 460.52 ha. Four are privately owned forestland plantations where logging and bamboo cutting are their main business activities. The Taiwan Forestry Research Institute, a governmental entity, operates the fifth certified forestland. This institute owns several natural forestlands preserved for the purpose of forest research and ecosystem services provision (see Table 3). This state-owned organization claims that no commercial logs are harvested, and these forestlands are managed only for research.

## 4. Material and Methods

Our empirical study targeted all five Taiwanese forest firms certified by the FSC. In-depth personal interviews and reviews of documents as well as field visits were administered to gather holistic information regarding business model transformation after certification. To encourage respondents to share their opinions and reduce ambiguity, a semi-structured questionnaire was designed by referring to previous literature and discussions with experts in the field of forest management, and was designed to suit local conditions [24,30,41,46,47]. This study selected participants based on their job responsibilities, positions, and involvement in FSC certification. A single interviewer conducted all interviews as a strategy to avoid interviewer bias. They were conducted on-site, with each interview lasting approximately one and half hours (please see Table 4 for more details). A semi-structured questionnaire with eleven open-ended questions covered value proposition, value creation and delivery, and value capture, with the aim of exploring the challenges related to sustainable business model innovation by certified forestry firms (see Table 5). Three questions were designed to find out if changes occurred with the firms' value propositions, six questions dealt with how these businesses aligned their activities with FSC standards and how they aligned their partnerships to channel resources to create and deliver sustainable value, and two questions targeted economic, environmental, and social value capture. After finalizing the survey, we reviewed the findings to cross-validate the interview results, on-site observations, and operational documents.

**Table 3.** Basic information of the five certified forestlands in 2016.

| Organization | Forest Activities | Forest Types | Number of Employees | Ownership/Forest Area (ha) | Annual Income ($US) | FSC Certificate Issued |
|---|---|---|---|---|---|---|
| Lienhuachi Research Center, Taiwan Forestry Research Institute, Council of Agriculture | forest management and forest research | Natural forest: Broad-leaved species (mainly Castanopsis, Cyclobalanopsis, Phoebe and Schima) Plantation: Cunninghamia lanceolata (Lamb.) Hook.; Eucalyptus citriodora; camaldulensis spp. | 33 | State-owned/460.52 | 0 | 2016-09-30 |
| Taiwan Leader Biotech Corp. | forest management and logging | Plantation: Softwood (Cryptomeria japonica); Hardwood (Machilus kusanoi; Trema orientalis) | 52 | Privately managed/58.02 | 0 | 2016-03-29 |
| Yong-Zai Forestry Co. Ltd. | forest management and logging | Plantation: Hardwood (Acacia spp.; Leucaena leucocephala; Swietenia mahogany, etc.) | 7 | Privately managed/385 | 266,666 | 2015-10-14 |
| Jang Chang Liang Co. Ltd. | forest management and logging and wood processing | Plantation: Softwood (Cryptomeria japonica; Taiwania cryptomerioides; Cunninghamia lanceolate) | 10 | Privately managed/212.15 | 264,662 | 2015-01-14 |
| Ai-Nun Enterprise Co. Ltd | bamboo management and logging | Plantation: Kuei Bamboo (Phyllostachys makinoi) | 44 | Privately managed/28.48 | 0 | 2017-03-23 |

Source: FSC Public Summary Report https://info.fsc.org/certificate.php#result.

**Table 4.** Interviewees information.

| Organization | Site Location | Position | Years of Experience (Approximate) | Interview Duration | Interview Location |
|---|---|---|---|---|---|
| Lienhuachi Research Center, Taiwan Forestry Research Institute, Council of Agriculture | Central Taiwan | Economic Department Head | 15 | 1.5 hours | Office |
| | | Economic Department staff | 10 | 1 hour | Office |
| | | Operation Department staff | 8 | 1 hour | On-site |
| Taiwan Leader Biotech Corp. | North Central Taiwan | General Manager | 11 | 1 hour | Office |
| | | Forest Management Manager | 5 | 2 hours | On-site |
| Yong-Zai Forestry Co. Ltd. | Southern Taiwan | Forest and Factory Manager | 5 | 1.5 hours | On-site |
| | | Forest Operation worker | 2 | 1 hour | On-site |
| | | Forest Management Planner | 2 | 1 hour | Office |
| Jang Chang Liang Co. Ltd. | North Central Taiwan | Owner | 40 | 1.5 hours | Office |
| | | Forest Management Manager | 24 | 1 hour | On-site |
| Ai-Nun Enterprise Co. Ltd | Southeastern Taiwan | Operation Manager | 3 | 2 hours | On-site |
| | | Agriculture Department staff | 5 | 1 hours | Office |

Source: recorded and organized by this study.

**Table 5.** Semi-structured questionnaire for interview.

| Elements of Sustainable Business Model | Questions | Rationale and Justification |
|---|---|---|
| Value proposition | What is the value of the products and services your firm is providing to the market? Has the value changed after the certification? What are the main reasons for providing this product and service value? | To explore whether the certified forest firms provide a sustainable value of products and services in concert with the triple bottom line. |
| Value creation & delivery | How did your firm implement FSC practices? What critical changes occurred in the changeover from your previous to your current business practices? What challenges did your firm experience while trying to implement FSC? How did you overcome these challenges? Name the people involved in your firm's changes in business practices. Describe the stakeholders involved in your firm's transformation. How does your firm deliver values to stakeholders? | To examine what activities need to be done when re-designing business operations for sustainability, how they implement them, and who performs the activities. |
| Value capture | Besides economic costs, are there any extra costs or impacts during the FSC implementation? What are the benefits earned by your firm after implementing FSC? | To comprehend whether the economic, environmental, and social value were captured by firms. |

Source: organized by this study.

To vigorously identify the causal effect of FSC adoption on sustainability transformation, this study applied a pattern matching logic to rule out alternative explanations that may have contributed to the transformation [48]. The matching process is done by proposing a predicted pattern (may have multiple relevant outcome variables) and comparing it with an empirically observed pattern. If two patterns match with each other, then the internal validity of the causal relationship is achieved. To validate our main proposition (pattern): forestry firms, by adopting forest certification, transform their conventional business models into sustainable business ones, we have the following multiple predicted outcomes:

➢ certified forestry firms propose the value of sustainable timber/non-timber products and ecosystem services to all stakeholders, including customers;
➢ certified forestry firms align with FSC standards and transform their activities, partnerships, and channels to create and deliver sustainable value;
➢ certified forestry firms promote not only economic values, but also environmental and social values.

If the above predicted outcomes matched with empirical evidence collected from interview results and onsite investigations, we successfully created a link between FSC adoption and sustainable business model transformation.

## 5. Results and Discussions

The following interpretations and analyses presented correspond to the three previously mentioned elements (value proposition, value creation and delivery, and value capture) of sustainable business models:

### 5.1. Value Proposition

To examine whether a certified forestry firm had transformed from its conventional to sustainable value proposition, this study gauged the changes in product and service value before and after certification, and, furthermore, sought to discover the motivations behind FSC adoption. From the responses of all interviewees, they unanimously responded that product and service values changed after joining forest certification. Before being certified, the firms' purpose (value proposition) was solely

to supply timber/non-timber products to customers. Now, the goal (value proposition) was to provide certified logs, non-timber products (e.g., mushrooms, etc.), and/or ecosystem services (e.g., eco-tourism, water source protection, etc.) from responsibly managed forestlands for the benefit of stakeholders (customers, workers, environmental/social groups, etc.) and communities. However, contrary to the mainstream concept in business models that firms ought to first change their value proposition before aligning their business practices to create and deliver this changed value, our empirical findings suggest otherwise: that transformation begins by adhering to FSC standards. In the process of adopting these procedural changes, business owners, executives, and even some lower-level staff began to realize they were pursuing different values than before. Such a discovery by these personnel is critical, since after the change in practice, less resistance would have been encountered in these changing organizations. Furthermore, stronger support from decision makers could be expected following the value change.

These firms also revealed the reasons why they adjusted their perceptions: (1) to divest themselves of a poor reputation, (2) to differentiate their products and services in the market, and (3) to rekindle public trust. This result is consistent with the views of forest owners in Slovakia who consider certification to be a tool to improve their corporate image [49]. Ever since the 1990s, Taiwanese residents have regarded logging as a major cause of landslides and ecosystem destruction.

> "[ . . . ] for the past 20 years, we have been accused of environmental destruction by environmental groups [ . . . ]". "[ . . . ] we don't like to be told that our logs were from unattended forestland [ . . . ]". "[ . . . ] we would like to provide certified logs to alter people's perceptions of our firm [ . . . ]".

Third party certification assures the general public that a forestry operation complies with the principles and criteria of managing forestland in a sustainable way [50]. Certification alters stakeholders' negative perceptions while earning public trust. Hence, empirical evidence supports the predicted outcome: certified forestry firms propose the value of sustainable timber/non-timber products and ecosystem services to all stakeholders, including customers.

While certified firms re-oriented their value proposition after adoption, we observed a subtle difference between the private and public forestland owners. Little economic benefit is obtained from forest operation for the private forest firms in our study, but the certification helps to change the image of entire organizations to the public, and potentially increases the popularity of the firms' non-forest products. With the expectation of future profit, the employees of all levels are motivated to work together during the transformation, which is reflected in the high frequency of communication among employees with different functional roles. Nonetheless, such a scenario did not appear in the public institution of our study. There is no economic motive for this public institution in our study; the demand for FSC adoption is from environmental groups and the public, which urge governmental institutions to take a leading role in sustainable forest management. In response to this demand, the institution head from this public institution commanded his employees to initiate the certification processes. The adoption was carried out by dividing the tasks required by external experts and assigning them to each employee judging by their predesignated functionality (external expertise was also necessary for this small public institution during the adoption), then these tasks were performed passively as additions to their work routines. Contrary to the private firms, no inter-employee collaboration was observed in this institution, in which we doubt there is a consensus among employees as to the value of sustainability.

## 5.2. Value Creation and Delivery

This section discusses the changes in value creation and delivery by comparing the differences in firms' activities, partnerships, and channels before and after certification. Difficulties encountered in the transformation process are also discussed.

Previous literature points out that the lack of forest certification education is a barrier for its adoption in Cameroon [38]. Similarly, managers of small Taiwanese forestry firms have admitted

that they had no idea about the meaning of sustainability before certification, not to mention how to implement it. Conventionally, the Taiwan Forestry Bureau has disseminated new technology or practices to private businesses in the forest industry, and this has been the dominant approach to upgrading forestry operations. Unfortunately, very few technicians in the bureau are intimately acquainted with sustainable forest management (i.e., the ability to strike a balance among economic viability, environmental protection, and social equity). The bureau's promotional ideas of ecosystem management are often incomprehensible because of insufficient engagements with forestry firms and confusing jargon. Furthermore, up-to-date sustainable forest management and certification information is inaccessible due to the absence of FSC branches in Taiwan.

Teece (2007) proposed that large forestry firms (e.g., public/national forest land holders) may conduct their own research and alter their conventional forest operations to meet the standards demanded by the FSC. Their established management systems and procedures possess strong dynamic capabilities in terms of sensing, seizing, and transforming, which allows them to adopt FSC independently without external support [51–53]. However, we found that small Taiwanese companies require external expertise during FSC adoption, since they often lack internal resources. The interviewees stated that they learned of the most recent (within the last five years) sustainable forest management information in seminars held by college researchers, and that these seminars offered them an opportunity to overcome their lack of knowledge and their technical barriers.

One forest owner stated:

"[ . . . ] what is the meaning of sustainability and forest certification? The government doesn't promote them, and we have no clue [ . . . ]". "[ . . . ] after participating in forest certification and sustainable forest management seminars hosted by college experts, we start to have a rough idea [ . . . ]."

Language can be another obstacle hindering the transformation to sustainable forest management. Almost all materials available are in English, whether in hard copy or on the web. This, of course, creates a barrier for non-English speakers in their attempts to comprehend the material presented. Furthermore, Alemagi et al. (2012) and Naussbaum et al. (2000) indicate that the documents and procedures available on forest certification are also highly technical, and this makes them even more difficult to read, assimilate, and implement [38,54].

During the interviewing process, one of the forestry firms recalled the situation before certification:

"[ . . . ] I am not sure who we can consult with [ . . . ]. they're (related documents) all written in English and on the internet. We need experts to instruct us what and how to adjust our current forest management practices [ . . . ]."

In sum, sufficient support from experienced consultants or academic researchers is critical before small forest firms can tackle the issue of sustainability. In some cases, the transformation of small firms is even easier than for large firms, due to their having fewer established assets and procedures [53].

Empirical results from our interviews and reviews of internal documents suggest that conventional practice focuses only on harvesting, silviculture, and bio-diversity protection activities. The major changes after the introduction of FSC standards is for adopting firms to devote resources to the management of broader social issues, such as the economic development of the local community, employee training and compensation, and the implementation of health and safety measures, etc. Furthermore, the FSC enables adaptive forest management. In other words, adoption may mean that organizations cultivate dynamic capabilities.

We have categorized the activities required by FSC's 'principles and criteria' into four major steps starting from planning and pre-accessing, to implementing and monitoring practices. Later in this section, we will discuss the changes corresponding to each of the steps.

Respondents stated that before certification, the government only required forest firms to submit a simple harvesting plan before carrying out logging activities. After certification, the first major change

was to design a complicated and adaptive forest management plan, which focused not only on the forestland itself, but also on the potential social and environmental impacts caused directly or indirectly by firms' harvesting activities. The plan, therefore, serves as a set of instructions for managers and workers to attain sustainable management goals with respect to harvesting plans, silviculture planning, endangered species protection measures, soil protection, invading species wide-spread prevention, pesticide controls, labor health and safety implementation, and local community development plans. As a forest owner expressed:

> "[ … ] after certification, we need to assess both the environmental value and social values before cutting logs. We care about our environment which includes bio-diversity and ecosystem, and social equity such as neighborhood villages, and healthy labor and safety [ … ]."

For small organizations, where the majority of employees are field workers, significant effort must be devoted to administrative tasks such as document preparation in the wake of management planning. This framing, writing, and planning of documentation creates tremendous stress for those involved.

The first step of the management plan is to pre-assess impacts before forestry activities occur. This pre-assessment involves gathering concerns from local communities, diagnosing forestland conditions, and estimating the growth rate of tree species. Firms can now execute actions to prevent, mitigate, and/or repair any negative impacts from forestry activities. This practice lessens environmental and social costs and minimizes operational risk. A major change at the implementing stage is to build long-term relationships with stakeholders, in particular with local communities influenced by the effects of firms' operations. Engaging people who have vested interests and allowing them to express concerns caused by forest activities enhances transparency (i.e., providing forestry activities and data online) and develops mutually beneficial relationships (i.e., helping local economies or protecting natural environments). Even though consulting with stakeholders and communities can prevent negativity and increase positive social impacts, the biggest challenge is that these targeted small-scale certified Taiwanese firms do not regard this move to sustainable forest practice as a critical step, but rather view it as an extra and tedious step, especially in the beginning.

Another significant change after FSC adoption relates to labor rights. Conventional models pay minimum attention to labor issues. The only guideline for executives dealing with these issues is to adhere to the minimum requirements as regulated by government; issues include management of wage rate negotiation, workers' complaints, and occupational safety insurance. This study observed a huge leap in labor rights after certification. These organizations are required to establish formal rules to process workers' issues, and are obligated to record all related activities to labor relationship management. The practices to enhance worker health and safety are fine-tuned after adoption. It is at this stage that formal administration regulations for employee safety make their debut, and when employee training is required to ensure the safety of workers.

FSC encourages the adopting firms to proactively engage with local communities and to try to prevent or remedy any potential damage resulting from forestry operations. Therefore, after careful assessment of the impact due to forestry activities, the second stage is to implement actions to improve the wellbeing of local communities. For instance, Taiwan Leader Biotech built a pipeline to secure the supply of clean water for its neighborhood community, and made donations to the church, so that not only physical needs, but also spiritual needs were fulfilled. As another example, Yong-Zai Forestry Co. Ltd. provided employment opportunities for local residents by setting up a wood processing plant onsite.

Environmental quality is also a major issue considered at the implementation stage. In addition to the conventional business model, which only focuses on the production of wood products, the focus is extended to non-wood products and the surrounding ecological system. When implementing its forest management activities, Yong-Zai Forestry Co. Ltd. took a further step in creating a buffer zone around a stream in its forestland to prevent potential landslides. Meanwhile, it identified and developed a

highly value-added agroforestry product, wild mushrooms; the harvesting of wild mushrooms in the forestland then provided job opportunities for locals.

The monitoring stage aims to receive instant feedback and helps to adjust a firm's operation in a timely manner. Frequent on-site monitoring allows forestry firms to adapt implementation plans as a result of unexpected detrimental changes in their environmental and social conditions, as well as the availability of new technology beneficial to their management activities. This signifies a huge leap for these forestry firms, since on-site monitoring has been rare in the absence of forest certification. Without certification, the only monitoring activities conducted by the majority of Taiwanese forestry firms were to observe the survivability of young trees approximately one year after planting. Certified forest firms have now established monitoring procedures and recorded monitoring results in order to practice adaptive forest management as advocated by a sustainable business model. However, the increase in monitoring frequency dramatically inflates operational cost and affects profit margin; it therefore presents a major challenge for certified firms.

One forest manager stated:

"[ . . . ] joining forest certification . . . there are now more jobs to do after we harvest our logs [ . . . ] we now need to monitor for landslides, record and store that information [ . . . ]."

Along with the significant changes at each of the four steps, communication among forest firms and their stakeholders is enhanced after certification. Information transparency is vastly improved; forestry firms' management plans, FSC compliance activities, and monitoring results are published periodically online. Furthermore, personal contacts with affected parties become more frequent, both to enhance their wellbeing and to prevent any conflicts from impeding firm operation. A major task for social entrepreneurship is to achieve sustainability transformation of the market and to improve society. Communication with forest firms' stakeholders and even societies serves as a tool for these niche pioneers to promote their products and the underlying sustainable value, which is critical for not only expanding their market share, but also for promulgating the idea of sustainability [40]. The above findings endorse the second predicted outcome: certified forestry firms align with FSC standards and transform their activities, partnerships, and channels to create and deliver sustainable value.

*5.3. Value Capture*

The third element of the sustainable business model, value capture, contains the economic, environmental, and social components. This section will analyze the impact of FSC adoption on each of these components.

A firm's financial returns represent economic value capture. Generally, better returns can be realized either by market share expansion or by higher markup. Empirical evidence suggests that certified products are unable to reap higher economic value in the Taiwanese market due to intense competition from other non-certified products. Fortunately, an improved product image through FSC certification has attracted a few niche customers, and has led to the possibility of opening up new market segments or of market expansion. Such scenarios match a study across 117 countries, and show that consumers' preference for certified wood product increases as their incomes grow. This indicates that certification improves firms' advantages by expanding their market share, and allowing them to maximize profit. However, there is little evidence to suggest that certified wood products enjoy a price premium over conventional products, and the benefits of certification may not be sufficient to cover the higher costs associated with the new standards, even with the premium [55].

One forest owner said:

"[ . . . ] we couldn't charge premium price for our certified logs because the market is filled with non-certified logs. The market is too competitive [ . . . ]". "[ . . . ]. Instead, we found a niche market. An architecture designer called me once and asked for certified products for the construction of public building [ . . . ]".

Social value capture is achieved by providing job opportunities and cultivating new agroforestry products to help the local economy and make a major contribution to local communities. Social value capture is also reflected in improved worker health and safety programs resulting from comprehensive employee training and safety protection measures. A reduced accident rate is the most eye-catching evidence of labor safety improvement. A formal procedure to handle employee complaints and understand their needs is an example of social value that is conducive to employee welfare improvement. The final component, environmental value capture, can be realized through minimizing the negative externality resulting from forestry operations, and through increasing environmental benefits delivered by an improved ecosystem. A comprehensive environmental protection program has been developed in cooperation with the Taiwanese forestry bureau. The program now covers the protection of rare and threatened species, the protection and restoration of the ecosystem, the protection of water resources, and landscape conservation, particularly with respect to the prevention of landslides. One manager stated:

"[ … ] forest certification label helps increase our firm's positive image on environment protection to the public. People are discussing about us joining the forest certification. Our company name is recognized and we have expanded in the market [ … ]".

"[ … ] after we joined the forest certification, we can now claim that our certified forest products are from well-managed forestlands, which provide benefits to our environment and local community [ … ]".

In sum, small-scale certified forestry firms are transformed into sustainable business models by adopting forest certification. From the analysis above, we found that their new models incorporated the TBL approach, initially introduced by Elkington in 1997 [56], and encompassed the proposed sustainable elements. The proposed sustainable business model contains three elements. The first element, value proposition, depicts the benefits of environmental and social values contained in highly differentiated timber/non-timber products. Certified forestry firms not only engage with customers, but also with other affected stakeholders by proposing sustainable products and service value to satisfy their needs. The second element refers to certified forest firms' use of resources, key operational activities, partnerships, and channels for creating and delivering sustainable value. Armed with knowledgeable people from external institutions, certified firms partner with customers and stakeholders to create and deliver economic values and enhance the social and environmental wellbeing of society. Value capture is the third and final element in the proposed sustainable business model. It gives rise not only to economic worth, but also to positive environmental and social values. In terms of economic value, charging a premium price may not be possible in the short-term for certified forest firms. However, the expansion of sales to niche market consumers has been observed in some cases. Furthermore, long-term economic value may be achieved by eliminating the operational risk from potential boycotts or regulatory burdens due to firms' rising images and name recognition. Environmental value is created by conserving biodiversity and protecting ecosystem services following the practice of new management planning, and social value is enriched by providing local job opportunities and helping the economic development of communities.

## 6. Conclusions and Implications

Building on Bocken et al.'s (2014) sustainable business model archetype, along with Osterwalder and Pigneur's (2010) and Joyce and Paquin's (2016) business model canvas, the authors proposed that sustainable business model innovation may be achieved by forestry firms with the adoption of FSC [24,28,30]. The empirical results from this study confirm that forest firms have achieved business sustainability by considering economic viability, social equity, and environmental protection simultaneously. The sample of small-scale Taiwanese forestry firms dedicated substantial efforts to transforming into a sustainable business model, since access to sustainable forest practices and forest

certification knowledge proved to be problematic and challenging. The results of this study suggest that the outside-in approach, such as with the adoption of FSC certification, is an ideal way to cultivate sustainable business model innovations for small forestry firms that lack internal resources, such as an R&D department and/or innovative personnel. Outsourcing knowledge or obtaining expertise from consultants and/or experts from academic or governmental institutions will be necessary when trying to adopt forest certification to achieve sustainable management.

Although both private and public organizations regard FSC adoption as an effective means to alleviate the tension between forest firms and environmental groups, and to further improve public image, the motives for adoption are intrinsically different. Employees from private firms are driven by potential economic profit, followed by the improved corporate image, whereas there is weaker motivation for public employees due to the lack of economic incentives, which resulted in a low degree of communication during the transformation processes. Public employees act passively, without the formation of consensus on the value of sustainability. Such a scenario is potentially detrimental to the public organization; we argue that any external shock may then compromise the result of certification, due to the nonexistence of organizational learning (organizational learning plays an indispensable role for an organization to achieve a sustainable competitive advantage. Crossan, Lane, and White (1999) propose a theoretical model to illustrate organizational learning as a four-step process: initiating, interpreting, integrating, and institutionalizing [57]. Among these four steps, the processes of interpreting and integrating require communication among individuals and groups. Without sufficient interaction, an organization would not be able to harvest the fruit of organizational learning). As a result, FSC certification will be embraced by private organizations as long as the market demand for environmentally and socially conscious products increases over time. Nonetheless, much effort should be devoted from the public sector to carrying out a full-scale transformation, perhaps through changes in government regulation and mandating the certification in forest operation. Furthermore, there should be workshops held in public organizations to promote the value of sustainability and encourage inter-employee communication during the adoption.

Another interesting observation is the learning-by-doing transformation process in forestry firms' business innovation. Contrary to the conventional business model innovation, where innovation begins from a change in value proposition and is followed by an alignment of value delivery and value capture, our empirical results suggest that for these certificated firms, the changes start from the value delivery and capture element, which is initiated by the requirement of a comprehensive management plan. However, after carefully complying with these new standards, forest owners and staff understand and learn to embrace the idea of sustainability, which in turn changes their value proposition. The results of this study imply that the elimination of language barriers, the knowledge of forest certification training, adaptive management planning, and monitoring of the system are critical for prospective forestland holders to change their business model to one that is sustainable.

Although this study of small-scale Taiwanese forestry firms gives us valuable insight on the sustainable business model innovation, there is an inevitable limitation due to data availability. Managers often use financial performance indicators to gauge the performance of their business models, therefore, to compare the performance of these indicators after the sustainable transformation is an objective reflection of the results of economic value capture [58]. However, for the majority of firms in our study, forest operation is not yet a major source of income to these firms, and no data is available to distinguish the forest operation income from other sources of income. In particular, for the only public organization in our study, the aim of this organization is to conduct forest research for the public with financial support from the government; the pursuit of economic profit is never considered in this organization. As a result, we did not perform any comparison of financial ratios in our study. The only indirect evidence related to economic performance is that the adoption of FSC helps to attract environmentally and socially conscious customers for these certified firms, which expands their current consumer bases.

**Author Contributions:** Conceptualization, J.-Y.L.; Writing—Original Draft Preparation, J.-Y.L.; Writing—Review & Editing, C.-H.C.; Visualization, C.-H.C.

**Funding:** This research received no external funding.

**Conflicts of Interest:** The authors declare no conflict of interest.

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
