# Peer review of "Efforts toward Creating a Sustainable Business Model: An Empirical Investigation of Small-Scale Certified Forestry Firms in Taiwan"

_sustainability, doi:10.3390/su11092523_

Reviewer 1 Report

The paper addresses a very interesting topic, that is, the evaluation of sustainable business models. Nevertheless, there are several points that deserve attention from the part of the author(s), namely, the need for reinforcing the literature review with references from the target journal and the positioning of the paper revealing the contribution (and novelty) for the literature on sustainable business models. The qualitative approach could be completed by providing further information on the 9 different components of each layer of the triple canvas business model approach. The implications are somehow limited. 

Reviewer 2 Report

Dear Authors,

thank you very much for your revision and for the explanations you made. Although I agree with most of your explanations, I would kindly ask you to at least address in the section of limitations and future development relevance of financial performance indicators mentioned in my former review as it is important to objectively assess the financial performance of the companies as well.

Your Reviewer 

Reviewer 3 Report

I think the authors have developed a solid work. This new version has been improved very much. Almost all the comments and queries formulated in the first revision have adequately been  responded. However, the authors should attend two comments before final acceptance

1) Maybe the section "Data Collection" must be changed by "Material and Methods" and "pattern matching" technique could be described in a better way

2) The first paragraph of conclusions must be placed in the Discussion Section

Author Response

Round  2

Reviewer 1 Report

The changes made are in line with recommendations previously displayed. Considering this, I recommend acceptance.